# An Optical Fiber Sensor Based on Fluorescence Lifetime for the Determination of Sulfate Ions

**DOI:** 10.3390/s21030954

**Published:** 2021-02-01

**Authors:** Liyun Ding, Panfeng Gong, Bing Xu, Qingjun Ding

**Affiliations:** 1National Engineering Laboratory for Fiber Optic Sensing Technology, Wuhan University of Technology, Wuhan 430070, China; 278242@whut.edu.cn (P.G.); xb2016@whut.edu.cn (B.X.); 2School of Materials, Science and Engineering, Wuhan University of Technology, Wuhan 430070, China; dingqj@whut.edu.cn

**Keywords:** optical fiber sensor, sulfate optical detection, fluorescence enhancement, phase-modulation fluorometry

## Abstract

A new optical fiber sensor based on the fluorescence lifetime was prepared for specific detection of sulfate ion concentration, where 1,1′-(anthracene-9,10-diylbis(methylene))bis(3-(dodecylcarbamoyl)pyridin-1-ium) acted as the sulfate fluorescent probe. The probe was immobilized in a porous cellulose acetate membrane to form the sensitive membrane by the immersion precipitation method, and polyethylene glycol 400 acted as a porogen. The sensing principle was proven, as a sulfate ion could form a complex with the probe through a hydrogen bond, which led to structural changes and fluorescence for the probe. The signals of the fluorescence lifetime data were collected by the lock-in amplifier and converted into the phase delay to realize the detection of sulfate ions. Based on the phase-modulated fluorometry, the relationship between the phase delay of the probe and the sulfate ion concentration was described in the range from 2 to 10 mM. The specificity and response time of this optical fiber sensor were also researched.

## 1. Introduction

Sulfate concentration as an evaluation index of crystalline corrosivity has been the focus of research in the field of concrete erosion [1]. There are many reasons for the sulfate ion concentration in water to increase, such as the dissolution of gypsum and other sulfate deposits, sulfite and thiosulfate oxidation in the air, and the discharge of domestic sewage and industrial wastewater. Sulfate ions can react with cement hydration products to precipitate into expansive crystals, which bring about expansion, cracking, and strength loss in concrete structures [2,3,4]. The construction life of concrete structures will be seriously affected, and there is a hidden danger of causing major accidents [5,6,7]. Therefore, the monitoring of sulfate ion concentration in a concrete environment has important engineering and economic significance for the early warning of durability problems of concrete structures.

As an ion probe, special modified gold nanoparticles can be used to detect sulfate ions specifically. They can detect the different concentrations of sulfate ions through the aggregation degree of gold nanoparticles [8] and the fluorescence intensity of a complex [9]. Raman spectroscopy has also been reported for the determination of sulfate ions dissolved in pore water of sediments [10], and it realized the diurnal variability change monitoring of the SO_4_^2−^ intensity of offshore seawater [11]. Other techniques have been adopted to determine sulfate concentration, such as titration [12,13], atomic absorption spectroscopy (AAS) [14,15], ion chromatography (IC) [16,17], spectrophotometry [18], atomic fluorescence spectroscopy (AFS) [19], and inductively coupled plasma atomic emission spectroscopy (ICP-AES) [20,21]. Optical fiber sensors have many advantages compared with those detection methods: energy savings, high sensitivity, strong resistance to electromagnetic interference, enabling continuous remote monitoring, compactness, flexible shape, and real-time detection capability [22,23,24,25]. Combining sensitive material with a fiber optic sensor is a significant research direction which could prepare a sulfate fiber sensor, which has great potential for practical detection.

For ion detection, most of the receptors belong to the off type (fluorescence quenching), which means that the fluorescence emission intensity decreases when the analyte is combined. Although the off type of detection is widely used, fluorescence enhancement (on) is preferable to quenching (off) because it reduces the chance of false positives, is more suitable for multiplexing, and could use multiple detectors simultaneously to produce unique responses to different analytes [26,27]. A substance that can specifically bind sulfate ions and produce a fluorescent reaction is a superior fluorescent probe whose name is 1,1′-(anthracene-9,10-diylbis(methylene))bis(3-(dodecylcarbamoyl)pyridin-1-ium) (AMDP) [28]. The probe AMDP can combine with sulfate ions through hydrogen bonds to form a fluorescent complex and realize the detection of sulfate ion concentration based on the fluorescence enhancement effect. According to the molecular orbital theory [29], the reduced fluorescence effect of anthracene moiety in the probe AMDP is due to electron transfer between the anthracene moiety and the pyridine ring, which means that the photo-induced electron transfer (PET) effect is generated and prevents the generation of fluorescence. The sulfate group can bind to the two pyridine rings on the probe AMDP through hydrogen bonding, which invalidates the PET effect in the complex and restores fluorescence [30,31].

Fluorescence lifetime refers to the average residence time of molecules in the excited state before returning to the ground state after being excited by light pulse. The fluorescence lifetime is generally absolute and only related to the microenvironment of the fluorophore. It has been reported that fiber-optic sensors exist based on fluorescence lifetime to detect temperature [32], strain [33], and Fe^3^^+^ ion concentration [34] in solution. As it effectively avoids the inherent characteristics of fluorophore photobleaching, spectral drift, and inappropriate excitation light, the optical fiber sensor based on fluorescence lifetime detection has high-precision and stable detection performance.

In this paper, a novel method based on fluorescence lifetime for detecting sulfate ions was proposed by combining a sulfate-sensitive fluorescent cellulose acetate (CA) membrane with an optical fiber sensor. To immobilize the probe AMDP, the immersion precipitation method was adopted to prepare the sulfate-sensitive CA membrane. Polyethylene glycol (PEG) 400 was used as porogen to improve the porosity of the sensitive membrane and enhance the detection ability. The lock-in amplifier was used in the optical fiber sensor to collect and convert the fluorescence lifetime signal into a phase delay, which avoids the photobleaching effect to improve the anti-interference and detection capability of the sensor. Based on the principle of the phase-modulation fluorometry, the relationship between phase delay and sulfate ion concentration was researched. The sensing characteristics of the optical fiber sensor were further studied, including detection range, sensitivity, repeatability, and selectivity.

## 2. Experimental Methods

### 2.1. Materials and Apparatus

All chemicals used were of analytical-reagent grade. Cellulose acetate, dimethyl sulfoxide (DMSO), polyethylene glycol 400, sodium phosphate dibasic, sodium bromide, sodium iodide, sodium chloride, sodium carbonate, sodium persulfate, sodium thiosulfate, anhydrous sodium sulfate, sodium nitrate, magnesium chloride hexahydrate, calcium chloride, and sodium hydroxide were obtained from Sinopharm Chemical Reagent Co., Ltd (Wuhan, China). 1,1′-(anthracene-9,10-diylbis(methylene))bis(3-(dodecylcarbamoyl)pyridin-1-ium) was procured from Heowns Biochem Technologies LLC (Tianjin, China).

The Fourier-transform infrared spectroscopy (FT-IR) spectrum of the fluorescent complex was obtained by a Nexus (Nexus-470, Thermo Nicolet, Thermo Fisher Scientific, Waltham, MA, USA) intelligent Fourier transform infrared spectrometer. The scanning electron microscope (SEM) photograph of the sensitive membrane was obtained using a scanning electron microscope (JSM-7500F, Jeol, Japan). A lock-in amplifier (SR-830, Stanford Research Systems, Sunnyvale, CA, USA) was used to convert the fluorescent signal into phase shift information. Ultraviolet–visible (UV–Vis) adsorption spectrum and fluorescence spectrum were obtained from a UV–Vis spectrometer (UV-2450, Shimadzu, Japan) and fluorescence spectrophotometer (F-4500, Hitachi, Japan), respectively.

### 2.2. Fluorescence Probe Detection of AMDP

First, 0.1 mM of standard fluorescent indicator solution was prepared by dissolving 1.07 mg fluorescent probe AMDP in 10 mL DMSO. Then, 0.1 mL of indicator solution was added into 3 mL of 1 mM sodium sulfate DMSO–H_2_O (3:7) mixed solution to detect the properties of the fluorescent complex. The optimal excitation wavelength and emission wavelength of the fluorescent complex were found by fluorescence spectra, and further discussed according to the UV–Vis adsorption spectra. The detection ability of AMDP for sulfate ion in different pH values was also studied. In addition, the luminescence time, fluorescence lifetime, complexation ratio, and photobleaching effect of the fluorescent complex were also investigated.

### 2.3. Preparation of Sensitive Membrane

Cellulose acetate sensitive membrane was prepared by immersion precipitation method [35,36,37]. DMSO was used as a solvent; 14 wt% cellulose acetate and 10 wt% PEG 400 were added to DMSO and stirred at room temperature for 6 h to dissolve. Then, 1.0 × 10 ^−4^ M fluorescent probe AMDP was added and stirred for 2 h to dissolve. Ultrapure water was used as a gel bath. The 3 mL casting solution was tiled in a petri dish with a diameter of 60 mm, and then immersed in gel bath to form a membrane. After 2 h of membrane formation, the membrane was taken out from ultrapure water and dried at 30 ℃ temperature. Then, the sensitive membrane was characterized by scanning electron microscope and infrared spectrometer. Leakage experiment was conducted to verify the reliability of the sensitive membrane. The sensitive membrane was immersed in 10 mM sulfate ion solution to restore fluorescence and keep for 96 h. During this period, the absorption spectra of the solution into which the sensitive membrane was immersed were studied.

### 2.4. Sulfate Ion Detection with the Optical Fiber Sensor

The optical fiber sensor monitored the sulfate ion concentration by detecting the change of fluorescence lifetime, which avoided photobleaching and undesirable light-source interference, thereby improving the reliability and accuracy of detection. The lock-in amplifier was adopted to collect fluorescence phase change data, and the schematic illustration of the optical experimental platform is shown in Figure 1. The bifurcated fiber bundle consisted of 37 identical plastic optical fibers, with a core diameter of 0.25 mm and a fiber cladding of 30 μm. The sensitive membrane was excited by an LED light source with emission wavelength of 410 nm at 40 kHz frequency through the bifurcated fiber. The fluorescence signal was collected by an optical fiber contact (FC) optical connector, modulated by the lock-in amplifier, and then transmitted to the computer for further processing and recording. The sensitive membrane was cut into a small wafer of appropriate size and placed in the metal probe for detection. After each concentration of SO_4_^2−^ ions, the membrane needed to be replaced with a new one in the sensor. The sensitive membrane was fastened in the sensor probe through a metal fixing nut, and the diagram is shown in Figure 1.

The sensitive membrane was immersed in the solutions of the different sulfate ion concentrations, and SO_4_^2−^ combined with the fluorescent probe AMDP on the membrane to form a fluorescent complex to emit fluorescence. The relationship between fluorescence intensity, fluorescence lifetime, and concentration of SO_4_^2^^−^ could be described by Equation (1) [34]:(1)$$\frac{I_{1}}{I_{2}}=\frac{\tau_{1}}{\tau_{2}},$$
where *I* is the fluorescence intensity and *τ* is the fluorescence lifetime of the sensitive membrane. In the formula, subscripts 1 and 2 mean data of the sensitive CA membrane with the different SO_4_^2−^ ion concentrations.

The fluorescence was emitted by the sensitive membrane via a 410 nm wavelength LED light source at 40 kHz frequency *(f)*, which could be converted into phase shift information (*φ*) by lock-in amplifier. The following equation shows the relationship between phase delay (Δ*φ*) and fluorescence lifetime [38,39]:(2)tanΔφ=2πfτ,
the following equation can be obtained by combining Equations (1) and (2):(3)$$\frac{tan\Delta \varphi_{1}}{tan\Delta \varphi_{2}}=\frac{\tau_{1}}{\tau_{2}}$$

According to Equation (3), we can determine the change of fluorescence lifetime by the change of phase delay.

The prepared sensitive membrane was cut into prototype sheets of uniform size for the detection process. The assembled optical probe was immersed in the DMSO-H_2_O (3:7) solution to detect its sulfate concentration to obtain the standard concentration curve and formula. According to the environmental requirements of real detection [4], the concentrations of sulfate in the solution were 0, 2, 4, 6, 8, and 10 mM.

There are many other anions in the concrete environment, so to detect the specificity of the optical fiber sensor, the experiment proceeded with the anions of HPO_4_^2^^−^, Br^−^, Cl^−^, CO_3_^2^^−^, I^−^, S_2_O_3_^2^^−^, SO_3_^2^^−^, and NO_3_^−^ under the same conditions as sulfate.

## 3. Results and Discussion

### 3.1. Characterization of the CA Membrane

SEM images were taken in order to explain the effect of PEG 400 on the morphology of the sensitive CA membrane. Figure 2a depicts SEM surface section image of sensitive membrane produced without PEG, and the surface micromorphology shows many micro-sized holes and pits. These holes and pits of the surface are not connected with the interior, as shown in the cross-section of Figure 2b. As the solvent of casting solution, DMSO was gradually replaced by ultrapure water when the casting solution was immersed in a gel bath, which led to cellulose acetate gradually precipitating on the membrane. At the same time, PEG 400 rapidly moved from the casting solution to the ultrapure water because of its good compatibility with ultrapure water, resulting in many pore channels left in the membrane. The addition of the porogen, PEG 400, obviously promoted the formation of pores on the surface of the membrane and made a three-dimensional pore structure form in the membrane, as shown in Figure 2c,d. When the sensor with a porous, sensitive CA membrane was used to detect sulfate solutions, the surface and internal pores of the membrane allowed the sulfate ions in solution to fully contact AMDP, which promoted the enhancement of the fluorescence of the sensitive membrane and thus the detection of the sensor. As shown in Figure 2c, the SEM image was processed, and the results showed that the average pore size of CA membrane was 0.574 μm.

The chemical structure of the fluorescent probe AMDP is shown in Figure 3a [28] and the FT-IR spectra of the CA membrane with and without the addition of AMDP are shown in Figure 3b. The CA membrane with AMDP has a series of characteristic peaks by comparison, such as 3089, 2928, 2856, 1664, 1589, 1501 cm^−1^. The absorption peak at 3089 cm^−1^ is the C–H stretching vibration peak of the aromatic hydrocarbon ring; the peaks at 1589 and 1501 cm^−1^ are both stretching vibration peaks of the aromatic hydrocarbon ring skeleton; the peaks at 2928 and 2856 cm^−1^ are respectively the C–H antisymmetrical and symmetrical stretching [40] vibration peaks of the alkane; the peak at 1664 cm^−1^ is the C=O stretching vibration peak of the amide [41]. Within the fingerprint region, the absorption peaks 845 and 675 cm^−1^ are the out-of-plane deformation vibration absorption peaks of C–H. Therefore, the AMDP was successfully fixed into the CA membrane by the immersion precipitation method and the composition structure of the probe was not changed.

### 3.2. Properties of the Fluorescent Complex

In order to verify that the probe AMDP can combine with sulfate ions to produce a fluorescent complex, we dropped the prepared the AMDP solution into a sodium sulfate solution, and obtained its excitation (EX) and emission (EM) spectra by fluorescence spectrometer. The EM spectrum of the mixed solution showed a separate fluorescence peak, which represented the successful generation and fluorescent emission of the AMDP–SO_4_^2-^ complex. The UV–Vis absorption spectra of the AMDP–SO_4_^2−^ complex showed that there were several characteristic absorption peaks of anthracene group in the range from 300 to 400 nm. Additionally, the peak values of the EX spectrum of the complex corresponded to the characteristic absorption peaks of anthracene group, which proved that the fluorescent cluster of the AMDP–SO_4_^2−^ complex is the anthracene group, as shown in Figure 4a,b. When the AMDP is bonded with sulfate ion via a hydrogen bond, the change of internal structure leads to the disappearance of the PET effect, which is also proved by the recovery of fluorescence of the anthracene group.

According to Equation (1), it is known that the increase of fluorescence lifetime is proportional to the increase of fluorescence intensity under ideal conditions, which indicates that the higher the fluorescence intensity is, the more significant the detection of fluorescence life will be. In order to improve the detection accuracy of fluorescence lifetime, the excitation wavelength of LED used in the sensor should be the optimal excitation wavelength of fluorescent complex. Figure 4c represents that the excitation wavelength of LED at 390 nm can achieve the best fluorescence lifetime detection effect.

In addition, the luminescence effect of the fluorescent complex was different under the different pH value environments. With the increase of OH^−^ ion concentration, the ability of the AMDP probe to bind sulfate ions with hydrogen bonds is inhibited, leading to a rapid decline in the detection ability under the pH environment close to 13. The cement starts facing corrosion issues when the pH value is higher than 10.5 [42], and the probe could work well in this environment, as shown in Figure 4d.

### 3.3. Response of the Fluorescent Probe when Detecting Sulfate Ions

There are three stages in the formation of the fluorescent complex by the reaction of the probe AMDP with a sulfate ion. The first 15 min are the first stage of the reaction. The first step is the rapid reaction of the probe AMDP with sulfate radicals to form fluorescent complexes; the fluorescence intensity of solution increases rapidly as a consequence. With the eventual decrease in reaction rate, the fluorescence intensity of the solution stabilizes at a plateau stage. The fluorescence intensity of the complex is stable from 15 to 25 min. When the reaction is completed, the fluorescence intensity of the fluorescent complex in the solution fluctuates for a period of time, and finally weakens to a very low fluorescence level. The fluorescence intensity of the complex at 25 min was used as the reference standard for measurement. The fluorescence intensity time curve of the fluorescent complex formed by the probe AMDP and sulfate ion is shown in Figure 5, and the dividing points of three stages are 15 min and 25 min, respectively.

### 3.4. The Complexation Ratio and Fluorescence Lifetime of the Fluorescent Complex

The complexation ratio of the fluorescent complex formed by the combination of sulfate ions and AMDP was researched by the Job plot method [43,44], as shown in Figure 6a. This result was confirmed by the Job plot method in which the fluorescence intensity exhibits a maximum at a molar fraction of approximately 0.5, indicating that a 1:1 stoichiometry was the most likely binding mode for AMDP and SO_4_^2−^ ion. The 1:1 binding ratio of the fluorescent complex was also consistent with the binding pattern mentioned in the references [26]. Fluorescence lifetime is the essential parameter of matter producing a fluorescent signal. It refers to the time required for the fluorescence intensity of matter to decay to 1/e after being excited, which reflects the average stagnation time of an electron in an excited state. The fluorescence decay curve of the AMDP–SO_4_^2−^ fluorescent complex was measured by time-correlated single photon counting (TCSPC), as shown in Figure 6b. The decay of the fluorescence lifetime of the complex was in accordance with the double exponential fitting calculation. The fluorescence lifetimes were 23.22 ns and 165.97 ns, and proportions of preexponential factors of both components were 27% and 73%, respectively. The period of the excitation LED source (40 kHz), 25 μs, was much longer than the fluorescence lifetime of the fluorescent complex, which means that the phase difference caused by the change of fluorescence lifetime will not be particularly obvious.

### 3.5. Reliability of the Sensitive Membrane

As an essential parameter of the sulfate ion optical fiber sensor, the reliability of the sensitive CA membrane was demonstrated. As shown in Figure 7a, the 10 mM sulfate ion solution was tested by the sensor and the phase was stable after 25 min, which showed that the fluorescent complex was successfully synthesized in the sensitive membrane and the stability was excellent. A leakage experiment was also carried out to ensure that AMDP would not leak during the monitoring process and contaminate the test solution. The absorption spectra of the 10 mM sulfate solution into which the sensitive membrane was immersed from 2 h to 96 h were studied, as shown in Figure 7b. The characteristic absorption peaks of the fluorescent complex were not observed in the absorption spectra, which indicated that the immobilization of AMDP in the membrane was successful and that the fluorescent complex was stable during the measurement process.

### 3.6. Detection of Sulfate Ions with the Optical Fiber Sensor

To improve the reliability of detection, it is necessary to replace the old sensitive membrane before using this sensor to detect the concentration of sulfate ions. When the optical fiber sensor detected the sulfate ions, the fluorescent phase shift (*φ*) increased with the sulfate concentration, which was due to the different degree of fluorescence enhancement produced by the fluorescent probe AMDP and the sulfate concentration, as shown in Figure 8a. As the sulfate concentration increased from 2 to 10 mM, the fluorescence intensity and lifetime of the sensitive membrane also increased, and the sensor converted the fluorescence lifetime into a phase shift to obtain phase delay.

Due to the slight phase delay (Δ*φ*) of the sensitive membrane, *tan*Δ*φ* could be converted into Δ*φ* during calculation. Therefore, the mechanism of sulfate detection was based on the fluorescence enhancement effect, which could be described by
(4)$$\Delta \varphi =\Delta \varphi_{0}+K\left[Q\right],$$
where *K* and [*Q*] are the enhancement constant of the fluorescent complex and the concentration of SO_4_^2−^, respectively. In the formula, Δ*φ_0_* means the phase delay caused by mechanical error.

The average of the phase values was collected at the 1st minute and the 25th minute. The delay phase was obtained by the two average values, which could eliminate mechanical errors and obtain more accurate values. Each concentration of sulfate ions was measured by the sensor many times, and the corresponding standard deviation of phase difference value was obtained. The sensor has high detection stability, and the maximum variance and average variance were 0.00955 and 0.006657, respectively. The calibration experiment of the sensor revealed the relationship between the sulfate concentration and the phase delay, as shown in Figure 8b. The relationship between the phase delay (Δ*φ*) and the sulfate concentration followed Equation (4), and a linear relationship equation $\Delta \varphi =0.0091+0.01396\left[SO_{4}^{2-}\;\right]$ (R^2^= 0.99553) was obtained for a sulfate concentration range from 2 to 10 mM.

### 3.7. Selective Experiment of the Optical Fiber Sensor

According to a previous study [28], it is known that the structure of the fluorescent probe AMDP has a cavity suitable for forming hydrogen bonds with a sulfate ion, which means that it can specifically bind with a sulfate ion. We carried out selective detection of common anions (HPO_4_^2−^, Br^−^, Cl^−^, CO_3_^2−^, I^−^, S_2_O_3_^2−^, SO_3_^2−^, NO_3_^−^, SO_4_^2−^) in a concrete environment. The concentration of chloride ions was 1 M and other ions were 10 mM. The phase delay of the sensitive membrane (Δ*φ*) did not change significantly except for SO_4_^2−^ in the presence of each of the above chemical substances, which indicates that the optical fiber sensor has favorable selectivity for SO_4_^2^^−^ ions, as shown in Figure 9.

In addition, some typical interference cations in concrete, such as Ca^2+^ and Mg^2+^, have little impact on sulfate detection of the sensor compared with Na^+^, as shown in Table 1.

## 4. Conclusions

A new optical fiber sensor was prepared for detecting sulfate ion concentration based on the fluorescence lifetime, where AMDP contained in the sensitive membrane can specifically combine with a sulfate ion to form a stable fluorescent complex. The sensitive CA membrane with PEG 400 as the porogen has a porous structure on the surface and inside, which allows the sulfate ions in the solution to be fully combined with the fluorescent probe AMDP. The phase delay of fluorescence changed as the sulfate concentration increased from 2 to 10 mM, and the calibration equation of the sensor was $\Delta \varphi =0.0091+0.01396\left[SO_{4}^{2-}\right]$ (R^2^ = 0.99553, under DMSO-H_2_O (3:7) solution). This optical fiber sensor provided a promising method with repeatability, a short reaction time, and high selectivity. Moreover, the sensor has the potential for further research and development, and the hydrophilicity of the sensor can be improved by modifying the sensitive material to adapt to the detection environment.

## Figures and Tables

**Figure 1 sensors-21-00954-f001:**
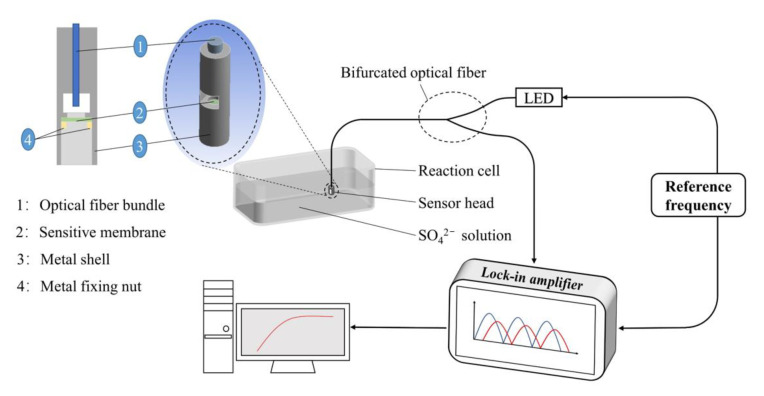
Schematic diagram of the optical fiber sensor.

**Figure 2 sensors-21-00954-f002:**
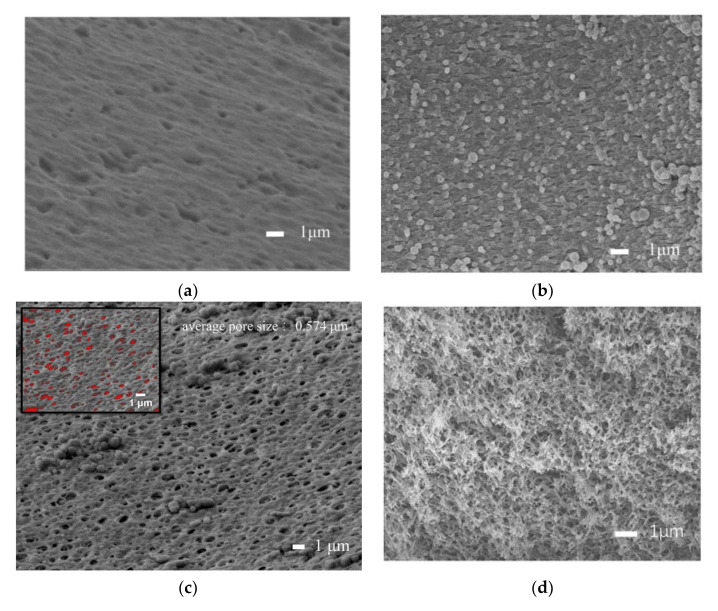
SEM images of sensitive CA membrane. Surface section (**a**) and cross-section (**b**) of the membrane without PEG 400; surface section (**c**) and cross-section (**d**) of the membrane with PEG 400.

**Figure 3 sensors-21-00954-f003:**
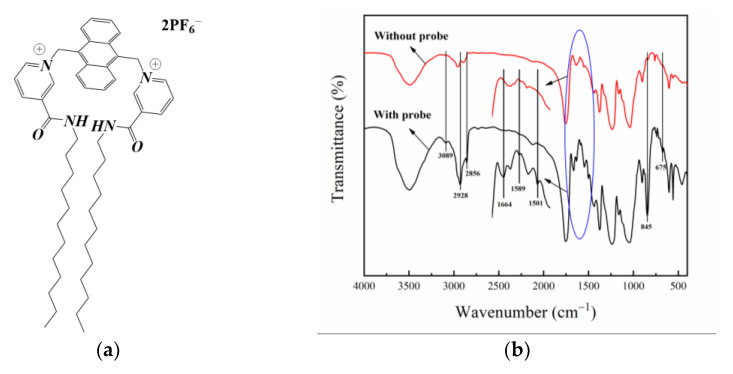
The structure of the fluorescent probe AMDP, (**a**) and the FT-IR contrast spectra of the CA membrane with and without the probe (**b**).

**Figure 4 sensors-21-00954-f004:**
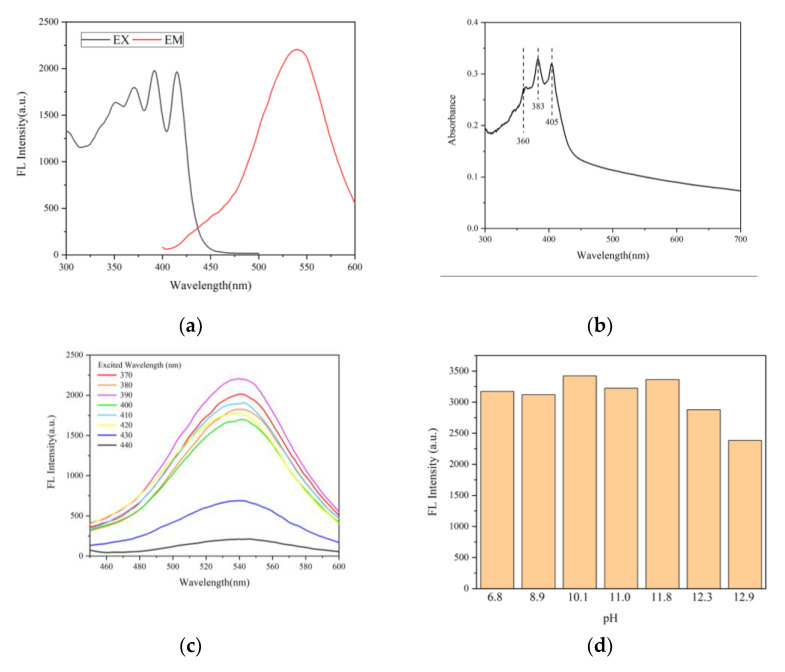
The fluorescence spectrum of the fluorescence intensity at 535 nm at different excitation wavelengths and the emission fluorescence spectrum at an excitation wavelength of 390 nm (**a**). Absorption spectrum of the AMDP–SO_4_^2−^ fluorescent complex (**b**). The fluorescence spectrum of the fluorescent complex at excited wavelengths from 370 to 440 nm (**c**) and the intensity of the fluorescent complex’s fluorescence under different pH conditions (**d**).

**Figure 5 sensors-21-00954-f005:**
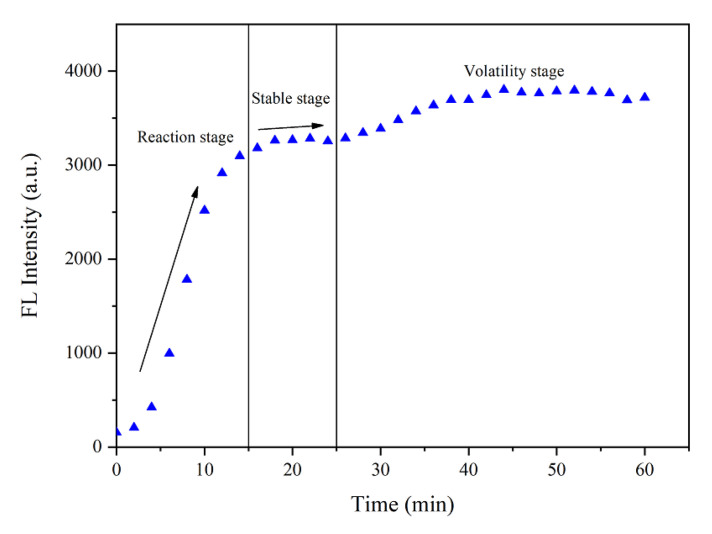
Time–intensity graph of the fluorescent complex.

**Figure 6 sensors-21-00954-f006:**
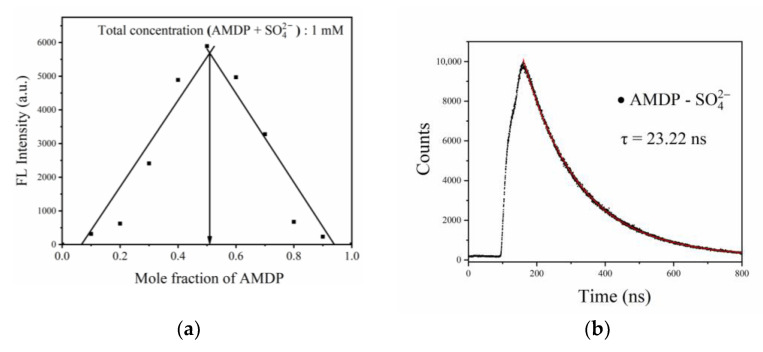
Fluorescence intensity diagram of the fluorescent complex with different ratios (**a**) and fluorescence decay curve of the fluorescent complex (**b**).

**Figure 7 sensors-21-00954-f007:**
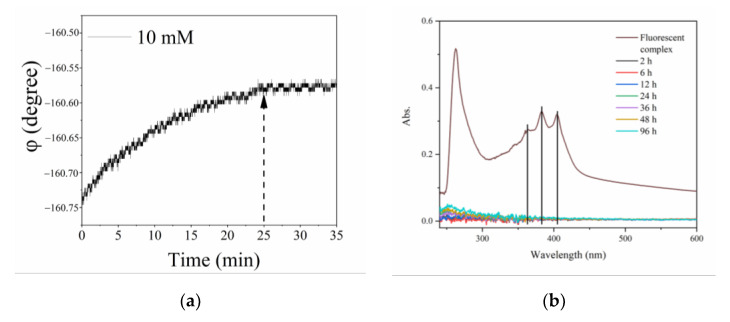
Time-phase diagram of sensitive CA membrane at detecting 10 mM sulfate ion (**a**) and the absorption spectra of the fluorescent complex and the sensitive CA membrane immersed for 2, 6, 12, 24, 36, 48, and 96 h (**b**).

**Figure 8 sensors-21-00954-f008:**
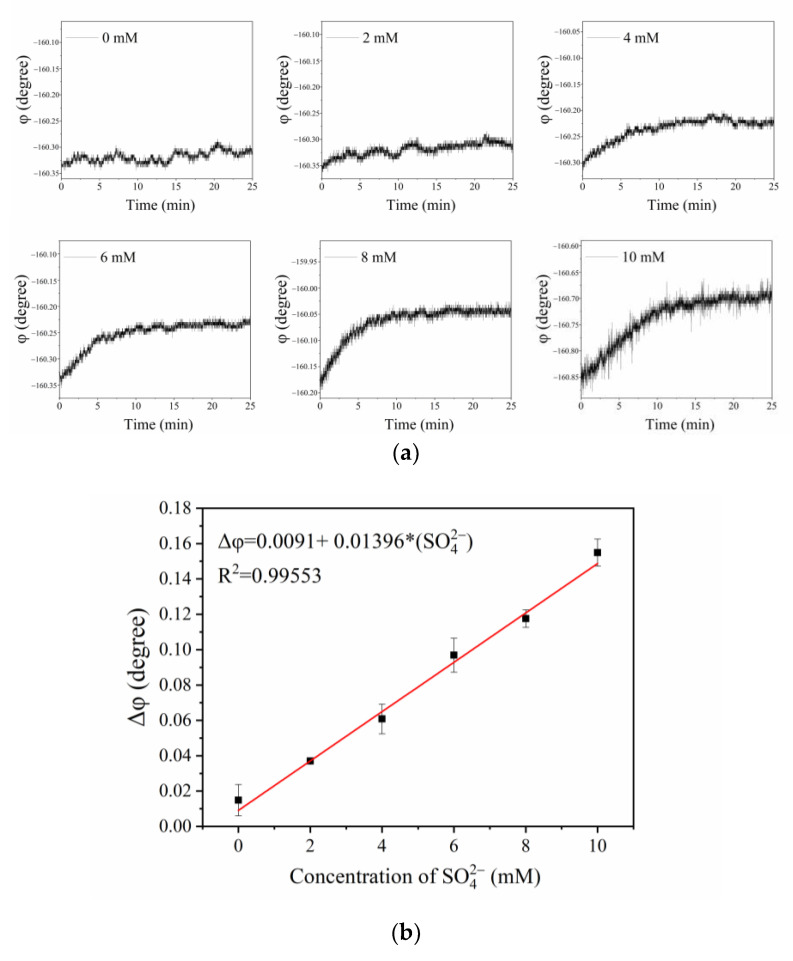
Time-phase diagram of the sensitive CA membrane at different sulfate ion concentrations (**a**) and the relationship between the phase delay (Δ*φ*) and the concentration of SO_4_^2−^ (**b**).

**Figure 9 sensors-21-00954-f009:**
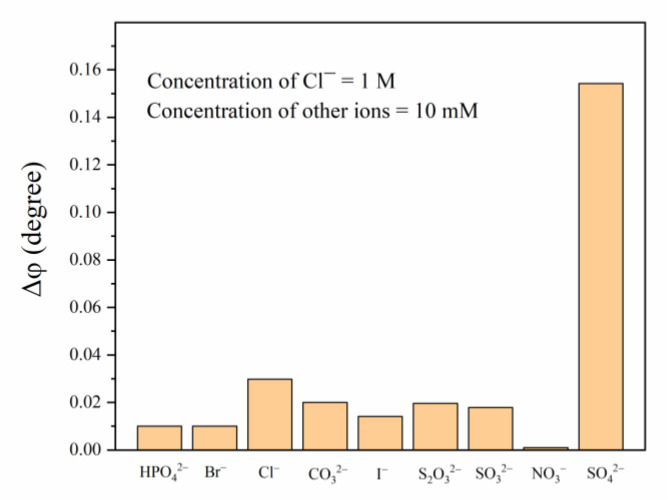
Effects of different ions on the phase delay (Δ*φ*) of the sensitive CA membrane.

**Table 1 sensors-21-00954-t001:** Influences of different cations on the detection of 10 mM sulfate ion by the sensor.

Cations	Δ*φ*	Difference
Na^+^	0.1543	
Ca^2+^	0.1544	+0.0001
Mg^2+^	0.1493	−0.005

## Data Availability

Data will be made available on request.

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
