# Peer review of "An Optical Fiber Sensor Based on Fluorescence Lifetime for the Determination of Sulfate Ions"

_sensors, 2021, doi:10.3390/s21030954_

Round 1
Reviewer 1 Report
The proposed manuscript pertains an interesting approach. An optical fiber-based sensor is fabricated and tested to detect sulfate ions using fluorescence life-time. The sensing structure is porous, both on the surface and inside, is used to probe for sulfate ions which diffuse and combine with the sensing platform forming a stable fluorescent compound that is then used to gage the concentration of sulfate ions.
The research proposed is complete and the data analysis is adequate. I would suggest that the author(s) add a few lines discussing the accuracy of measurements (discuss figure 7b error bars), may be calculate the average accuracy of the measurements for the used sulfate ion concentration domain.
The conclusions are reasonable and well corroborated by reported experimentation in the manuscript, but please make it clear in the manuscript how the sensor offers repeatability and to what extent (how many uses?)
The English style of the manuscript is of very low quality. There are dozens of grammatical errata and syntax errors as well.
Consider enhancing the quality of the figures, and increasing the font sizes for the axes labels. For example Figure 7: the time phase figures at very small and the axes labels are hardly visible.
Reviewer 2 Report
Manuscript No: sensors-1042851
Title: An optical fiber sensor based on fluorescence lifetime for the determination of sulfate ion
Authors: Liyun Ding, Panfeng Gong, Bing Xu and Qingjun Ding
- Overview
- In this manuscript the authors report on experimental research on optical fiber sensor for the determination of sulfate ion.
- The contents are expressed clearly; the manuscript is well organized and written in reasonable English.
- The authors have acknowledged recent related research.
- As long as my knowledge, the work presented is original and it is correct from a scientific point of view.
- Detailed analysis
Abstract: is too long. The abbreviations in the abstract are not needed.
- Introduction: provides an interesting approach to the subject and there are up to date references.
- Line 60, “molecular orbital theory” needs a reference
- Experimental methods
- Explain to readers why PEG 400 is a porogen.
- what is the meaning of AR ?
- Line 108: “PEG 400 was used as a porogen” is repeated several times throughout the manuscript
- Preparation of sensitive membrane and deposition on the optical fiber must be explained (also use a figure)
- Results and discussion
- A SEM image of the membrane with and without must be presented
- the authors must present result for membrane without PEG.
- Overall assessment
The work reported presents great utility for supplementary studies and developments in the field.
In my opinion it can be published after major corrections.
- Review Criteria
- Scope of Journal
Rating: High
- Novelty and Impact
Rating: High
- Technical Content
Rating: High
- Presentation Quality
Rating: High
Reviewer 3 Report
1-The English of the manuscript, in some sections, is very weak. It must be substantially improved. For example:
Lines 99-104: Please correct the „Dissolved 1.07 mg fluorescent probe AMDP in 10 mL DMSO solution to make…” e.g. to “1.07 mg fluorescent probe AMDP was dissolved in 10 mL DMSO solution to make 0.1…”. This happens repeatedly in this section e.g. „ Added 0.1 mL indicator…” … Please correct all of them.
Lines 110-111: “Ultrapure water as a gel bath, taken 3 mL of the casting solution in a petri dish and placed it in a gel bath to form a membrane” the sentence is not grammatically correct.
….
2- In formula 1, for the correlation of fluorescence lifetime and fluorescence intensity, please provide suitable references.
3-For equation 2 and 3: please provide the suitable references in which these equations have been used or more discussed.
4-Line 144: Why is DMSO-H2O (3:7) used as solvent? Does it match to a real sample?
5-Line 146: The evaluated concentrations are too high, especially for a fluorescence method. What are the concentrations in real samples e.g. concrete environment, which must be detected?
6-Lines 122-124 and 183-187 and figure 3: Timing is a very important factor in your developed sensing concept. “The sensitive membrane was excited by an LED light source with emission wavelength of 410 nm at a certain frequency through the bifurcated fiber.” What is this frequency? In figure 3, the time range is min (e.g. 10 -60 min). Please describe more in detail the timing, which is used for final measurements.
7-Section “Characterization of the CA membrane” can be mentioned as the first section at the very beginning of the Results and discussion.
8-Line 222: “…sensitive membrane soaked for 2 hours to 96 hours were studied,…”. Soaked in what?
Reviewer 4 Report
The paper developed a fluorescence lifetime based optical fiber sensor for monitoring sulfate ion in cement, which is beneficial for early prediction of concrete structure failure. It is recommended for publication after consider the comments:
Line 34 Line 55. “… to different analytes” please add citation
Line 65. “… restores fluorescence” please add citation
Line 76. Please give the full name and abbreviation at the first time in Line 74
Line 99. I noticed that the sentences are lake of subjects, reads like an experimental report rather than a published paper. Please write like “1.07 mg fluorescent probe AMDP was dissolved in 10 mL DMSO solution to make … indicator solution”, the problems are also found in other places such as Line 110 to Line 111
Line 114. Please give some details of leakage experiments. And also give detailed results of leakage experiments in Line 221 “…carried out to ensure that AMDP were decently immobilized” is so vague, readers get nothing from the leakage experiments
Line 143. The membrane was cut into sheets? Do you mean after each concentration of SO4 ion, the membrane needs to be replaced with a new one? If so, after each replacement, how to make sure the membrane is in the same position, considering optical sensing is very sensitive to position change
Line 172. You may not need to worry much about the high pH inhibition. In this literature, it points out that cement is stable at high pH, it starts facing corrosion issue when pH is less than 10.5. Your sensor works the best at this pH level.
Fei Lu, Ruishu Wright, Ping Lu, Paul R. Ohodnicki, "Metal oxides based fiber optic pH sensors for elevated temperature sensing applications," Proc. SPIE 11416, Chemical, Biological, Radiological, Nuclear, and Explosives (CBRNE) Sensing XXI, 114160P (26 May 2020); doi: 10.1117/12.2558088
Line 201. Please make a pore size distribution diagram using software such as ImageJ
Line 202 and Figure 5a. Citation of the chemical structure
Line 205-211. Citations of the chemical structure to each stretching vibration peak
Line 252. Are these ions in the same concentration as SO4 ion (10 mM, which you need to point it out)
Line 255. … has favorable selectivity “to SO4 ion” (I cannot type correct format of SO42-)
General comments (you may not need to make any changes)
- There is slight difference between cement and concrete, concrete is combination of aggregates (sand, gravel, crushed stones, …) and cements, where cement works as the paste. For monitoring chemicals in the concrete, it is generally means monitoring the chemical ions in the cement pores.
- For sulfate ion detection using optical fiber sensor, do you have any consideration about how do you deploy the probe in the concrete/cement? And when? During the mudding of the cement or after the cement is hydrated/solidified?
Round 2
Reviewer 2 Report
Manuscript No: sensors-1042851 REV2
Title: An optical fiber sensor based on fluorescence lifetime for the determination of sulfate ion
Authors: Liyun Ding, Panfeng Gong, Bing Xu and Qingjun Ding
- Overview
- In this manuscript the authors report on experimental research on optical fiber sensor for the determination of sulfate ion.
- The contents are expressed clearly; the manuscript is well organized and written in reasonable English.
- The authors have acknowledged recent related research.
- As long as my knowledge, the work presented is original and it is correct from a scientific point of view.
- Overall assessment
The work reported presents great utility for supplementary studies and developments in the field.
In my opinion it can be published.
- Review Criteria
- Scope of Journal
Rating: High
- Novelty and Impact
Rating: High
- Technical Content
Rating: High
- Presentation Quality
Rating: High
